

# Soil bacterial biodiversity is driven by long-term pasture management, poultry litter, and cattle manure inputs

Yichao Yang[1],[*], Amanda J. Ashworth[2],[*], Jennifer M. DeBruyn[3], Cammy Willett[1], Lisa M. Durso[4], Kim Cook[5], Philip A. Moore, Jr.[2] and Phillip R. Owens[6]

[1] Department of Crop, Soil, and Environmental Sciences, University of Arkansas at Fayetteville, Fayetteville, AR, United States of America
[2] Poultry Production and Product Safety Research Unit, United States Department of Agriculture, Agricultural Research Service, Fayetteville, AR, United States of America
[3] Department of Biosystems Engineering & Soil Science, University of Tennessee, Knoxville, TN, United States of America
[4] Agroecosystem Management Research Unit, United States Department of Agriculture, Agricultural Research Service, Lincoln, NE, United States of America
[5] Bacterial Epidemiology and Antimicrobial Resistance Research Unit, United States Department of Agriculture, Agricultural Research Service, Athens, GA, United States of America
[6] Dale Bumpers Small Farms Research Center, United States of Agriculture, Agricultural Research Service, Booneville, AR, United States of America

[*] These authors contributed equally to this work.

Corresponding author
Amanda J. Ashworth,
Amanda.Ashworth@ars.usda.gov

## ABSTRACT

Soil microorganisms are important for maintaining soil health, decomposing organic matter, and recycling nutrients in pasture systems. However, the impact of long-term conservation pasture management on soil microbial communities remains unclear. Therefore, soil microbiome responses to conservation pasture management is an important component of soil health, especially in the largest agricultural land-use in the US. The aim of this study was to identify soil microbiome community differences following 13-years of pasture management (hayed (no cattle), continuously grazed, rotationally grazed with a fenced, un-grazed and unfertilized buffer strip, and a control (no poultry litter or cattle manure inputs)). Since 2004, all pastures (excluding the control) received annual poultry litter at a rate of 5.6 Mg ha$^{-1}$. Soil samples were collected at a 0–15 cm depth from 2016–2017 either pre or post poultry litter applications, and bacterial communities were characterized using Illumina 16S rRNA gene amplicon sequencing. Overall, pasture management influenced soil microbial community structure, and effects were different by year ($P < 0.05$). Soils receiving no poultry litter or cattle manure had the lowest richness (Chao). Continuously grazed systems had greater ($P < 0.05$) soil community richness, which corresponded with greater soil pH and nutrients. Consequently, continuously grazed systems may increase soil diversity, owing to continuous nutrient-rich manure deposition; however, this management strategy may adversely affect aboveground plant communities and water quality. These results suggest conservation pasture management (e.g., rotationally grazed systems) may not improve microbial diversity, albeit, buffer strips were reduced nutrients and bacterial movement as evident by low diversity and fertility in these areas compared to areas with manure or poultry litter inputs. Overall, animal inputs (litter or manure) increased soil microbiome diversity and may be a mechanism for improved soil health.

# INTRODUCTION

Grasslands are the largest agricultural land-use category in the US, with 265 million hectares being used for grazing (*Bigelow & Borchers, 2017*). A continuously grazed system (CG) is defined as animals grazing pastures for extended periods without allowing plants to recover (*Natural Resources Conservation Service , NRCS*). In hayed (H) systems, forages are mechanically removed throughout the growing-season. Rotationally grazed (R) systems consist of strategically rotating livestock through paddocks to maximize forage productivity. This practice improves soil health (*Pilon et al., 2017a*), water quality, and conserves natural resources (*USDA-NRCS, 2019*). Another designated best management practice is the establishment of edge-of-field buffer or riparian buffer strips along water bodies. Riparian buffer strips decrease nutrient loading (*Lovell & Sullivan, 2006*; *Shearer & Xiang, 2007*).

Pasture management practices influence the soil nutrient status and consequently may affect soil bacterial communities. Overgrazing by livestock erodes soil (*Webber et al., 2010*; *Van Oudenhoven et al., 2015*). H systems improve soil quality by decreasing soil penetration resistance, bulk density, and improving aggregate stability and infiltration rates (*Cox & Amador, 2018*). Rotationally grazed pastures with a fenced riparian buffer (RBR) reduces sediment loss compared to continuously grazed practices (*Sanjari et al., 2009*). Further, RBR reduces erosion (*Pilon et al., 2017a*).

Poultry litter is a common nutrient source for pastures. It includes macronutrients (nitrogen (N), phosphorus (P) and potassium (K)), as well as trace elements (*Stephenson, McCaskey & Ruffin, 1990*; *Gerber, Opio & Steinfeld, 2007*). Poultry litter applied to pastures provides macronutrients and trace elements to soils, but repeated treatments may increase metals in soil (*Han et al., 2000*; *Moore Jr et al., 1998*; *DeLaune & Moore Jr, 2014*). Grazing also contributes to increased soil nutrients due to animal excreta (*Vendramini et al., 2007*). Nutrients from animal manure inputs (such as poultry litter and cattle manure) are also excellent microbial food sources (*Ashworth et al., 2017*). Trace elements are also required for bacterial growth and act as cofactors for essential enzymatic reactions in bacterial cells (*He et al., 2014*). Therefore, increased nutrients from poultry litter applications and grazing likely promotes bacterial richness and diversity, although their combined effects are unknown.

While the physical and chemical impacts of pasture management on soil properties are well understood, the long-term impact of common pasture strategies on the soil microbiome is less studied. The soil microbiome supports plant growth, carbon and nutrient cycling, and maintaining soil health (*Jousset et al., 2011*; *USDA-NRCS, 2019*; *Fierer, 2017*). However, whether grazing changes microbial function and diversity remains inconclusive. *Ford et al. (2013)* substantiated the claim that grazing affects the composition of soil microbial populations in grasslands via the variation in phospholipid fatty acid markers. Grazing may also reduce soil microbial biomass (*Chen et al., 2016*; *Zhao et al.,*

*2017*) and soil microbial diversity (*Olivera et al., 2016*). However, others have reported that moderate grazing increases soil bacterial community diversity (*Qu et al., 2016*).

In addition to management, soil pH, and soil moisture drives soil bacterial assemblages (*Fierer, 2017*; *Fierer & Jackson, 2006*). *Wu et al. (2017)* suggested there is a strong relationship between bacterial diversity and soil pH, with greater diversity occurring in basic soils and lower diversity being observed in acidic soils. Animal inputs (both poultry litter and cattle manure) may influence soil pH and soil N, which can modify the makeup of soil microbial community and diversity by altering the nutrient status (*Bardgett et al., 1997*). Moreover, in a continental-scale assessment of soil bacterial communities, *Fierer & Jackson (2006)* discovered that bacterial diversity is higher in neutral soils and lowest in acidic soils.

The present study used Illumina 16S ribosomal RNA amplicon sequencing to identify the relative abundance and diversity of bacterial taxa in soils following 13-years of conservation pasture management. Species diversity is the measure of both richness and evenness, and considers not only the number of species present but also how the species are distributed. The goal of our study was to assess shifts in soil bacterial community structure and diversity based on long-term pasture management. We hypothesize that conservation pasture management practices (i.e., RBR) will have a more diverse soil bacterial community. Conversely, we suspect H systems will have a lower richness and diversity, owing to reduced cattle manure inputs, with soils receiving no poultry litter (control) having the lowest diversity. The aim of this study is to identify soil microbiome community differences following 13-years of pasture management systems and identify how soil biodiversity is impacted by cattle manure and poultry litter deposition to ultimately inform best management impacts of pasture systems.

## MATERIALS AND METHODS

### Experimental design

A field study was initiated in 2004 at the USDA-ARS Dale Bumpers Small Farms Research Center in Booneville, Arkansas (N35°06′12″, W93°56′05″, 150 m altitude) to evaluate water quality affects from conservation pasture management (*Pilon et al., 2017a*; *Pilon et al., 2017b*; *Pilon et al., 2018*). Fifteen watersheds were constructed on a site with an average slope of 8% and on an Enders (fine, mixed, active, thermic Typic Fragiudults) and Leadvale silt loam (fine-silty, siliceous, semiactive, thermic Typic Fragiudults). Each watershed was 25 × 57 m for a total area of 0.14 ha. The dominant forage was common bermudagrass (*Cynodon dactylon* L.).

Three grazing management strategies (CG, H, and RBR) were implemented from 2004-2017 with three replications (Fig. 1). The H treatment was hayed three times annually (April, June, and October) to a height of 10 cm with a rotary hay mower (no cattle in these watersheds area). The CG watersheds were continuously grazed by one or two calves throughout the year (*Pilon et al., 2017a*). The RBR watersheds were rotationally grazed based on forage height. Three calves were placed in each RBR watershed when forage height was 20 to 25 cm and taken out when forage heights were 10 to 15 cm. The RBR

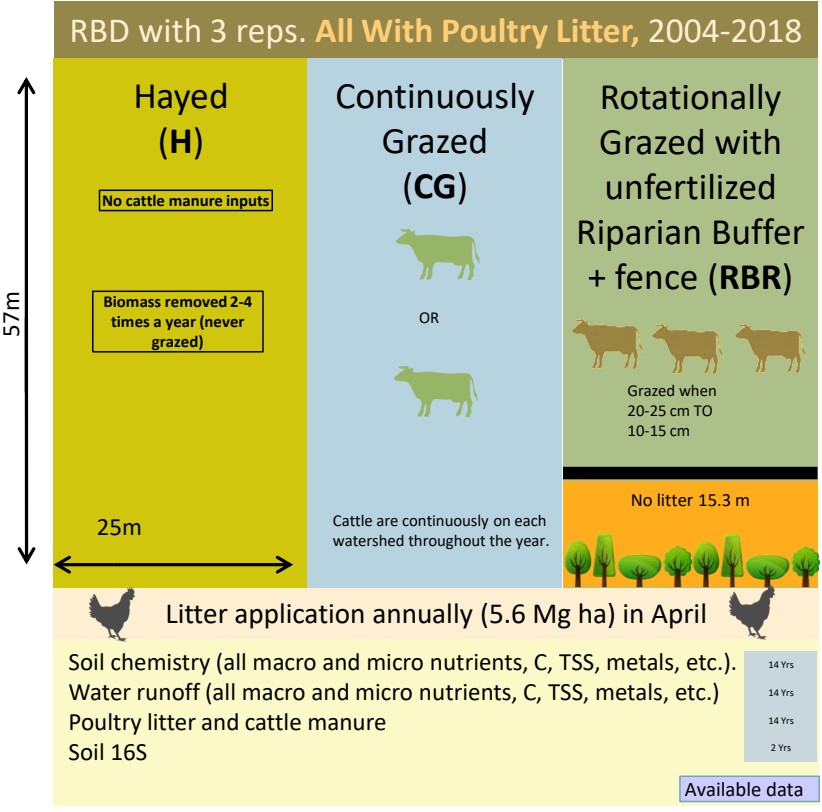

**Figure 1** **Schematic representation of the experimental set-up.** Randomized complete block design with three replications (nine watersheds total) from 2004–2018. All areas have received annual poultry litter applications (except for the control area). All watersheds received cattle manure excluding that of the Hayed treatment (H).

watersheds had a 15.3-m riparian buffer strip containing four tree species (i.e., white oak (*Quercus alba* L.), green ash (*Fraxinus pennsylvanica* Marshall), and pecan (*Carya illinoinensis* (Wangenh.) K. Koch) at the base of each watershed (Fig. 1). These areas were un-grazed, did not receive poultry litter, and were considered the control in this study (no poultry litter or cattle manure applied in this area). Each watershed was divided, perpendicular to the slope into three zones (corresponding to shoulder, upper backslope, and lower backslope positions), whereas the RBR consist of four zones (included the control; Fig. 1). Poultry litter was land applied at a rate of 5.6 Mg ha$^{-1}$ in April or May of each year to each watershed (excluding the control). Since poultry litter was omitted in the buffered area of the RBR treatment, application rates were identical on an aerial basis (RBR watersheds received 658 kg plot$^{-1}$, whereas H and CG received 794 kg plot$^{-1}$). Poultry litter was obtained annually from a commercial broiler farm near Booneville, AR.

## Soil sampling procedures and analysis

Soil sampling was performed every April and July during 2016 and 2017, once pre-poultry litter application and the other post-litter application (four sampling dates total). Soil

samples were collected from 0–15 cm and homogenized with a minimum of 6 randomly selected points in plot centers to preclude sampling borders, with three replicates total. To prevent contamination, soil was taken using probes sterilized between plots with 70% ethanol ($C_2H_6O$). Samples from each zone were collected and stored in a cooler for transport and stored at $-80°$ C for DNA extraction. Soil pH was measured with a pH electrode and conductivity meter on a subsample of the 1:10 soil extraction (SevenMulti, Mettler-Toledo). Total soluble nutrients (P, K, Ca, Mg) were determined by inductively coupled argon plasma spectrometry (Varian Vista-PRO), following a nitric-acid digestion using USEPA Method 3030E (*USEPA, 1979*).

## DNA extraction, PCR amplification, and sequencing

DNA was extracted from each soil sample using the PowerSoil DNA extraction kit (MoBio Laboratories) according to the manufacturer's directions. Briefly, the soil sample was fully homogenized, and 0.25 g of soil was taken from each sample for DNA extraction. Extracted DNA was quantified using Quant-It$^{TM}$ PicoGreen$^®$ (Invitrogen) dsDNA quantitation assay and stored at $-20$ °C.

Bacterial community composition was determined using Illumina Miseq sequencing of 16S rRNA gene amplicons. Extracted DNA was sent to the University of Tennessee Genomic Services Laboratory, where the V4 region of the 16S rRNA gene was amplified with barcoded primers 515F and 806R (*Caporaso et al., 2011*). Amplicon libraries were pooled and 291 base-paired end sequences were obtained on the Illumina MiSeq Platform, resulting in 15,172,724 total sequence reads. Reads were processed using the open source bioinformatics software Mothur V 1.40.0 following the Miseq SOP protocol (*Kozich et al., 2013*). After the quality control pipeline, 12,273,759 sequence reads remained using a 97% similarity threshold to define ribotypes in Mothur (19.11% were deleted).

## Data analysis and statistics

In this study, long-term pasture management (or the main effect; H, CG, RBR, and the control) was randomized, with zone or landscape position being the split-plot (based on slope position), and sampling timing (pre or post-poultry litter application) being the split-split plot treatment design. Prior to diversity analysis of soil microbial communities, the number of sequence reads in each sample were subsampled to 12,132 reads, the number of sequences present in the smallest sample, to eliminate effects from uneven sampling depth. At this size, sequence coverage for these libraries was good (0.97). The greengenes database was used to classify the operational taxonomic unit (OTU) at the genus level using the Bayesian method (*Cole et al., 2009*), thereafter relative abundance of all OTUs were summed within phylum and analyzed for relative abundance of OTUs at the phylum level. Based on this subsampled dataset, richness was calculated by using Chao index and diversity was calculated by measuring the inverse of Simpson using Mothur, and differences by treatments were analyzed by ANOVA in the statistical software R 3.5.1 (*R Core Team, 2012*) and JMP$^®$12 (*SAS Institute, 2014*). Bacterial community structure was quantified in a matrix of Bray-Curtis similarities, which was then analyzed in a permutational analysis of variance (PERMANOVA) to compare bacterial communities at the phylum

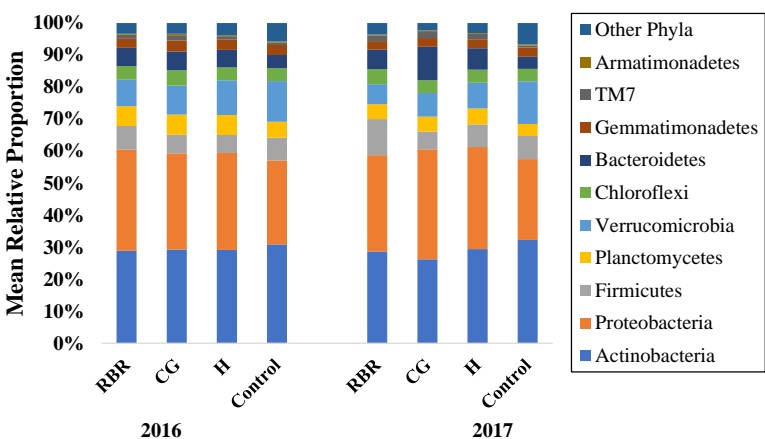

**Figure 2** **Mean relative proportion of soil bacteria phyla by treatment x year.** Pasture management includes continuously grazed (CG), hayed (H), and rotational grazed with a fenced riparian buffer (RBR). The RBR treatment consists of an additional non-grazed zone without poultry litter or grazing (control). The order of colors is the same in the legend as the bars.

level by fixed factors in PRIMER-E (*Clarke & Gorley, 2006*). Bacterial community structure was also quantified by using the ordination method of Principal Coordinates Analysis (PCoA) on a matrix of Bray-Curtis dissimilarities using MicrobiomeAnalyst (*Dhariwal et al., 2017*). Linear Discriminant Analysis Effect Size (LEfSe) measurement was used to identify taxa differences between treatments with Galaxy (*Segata et al., 2011*). Potential functional capacity of the soil bacterial communities were predicted using a random forest algorithm based on the classification performance on different treatments using MicrobiomeAnalyst (*Dhariwal et al., 2017*). MicrobiomeAnalyst was also applied to visualize the KEGG metabolic networks along with pathway analysis. It was used to predict possible effects of bacterial community composition shifts in function.

## RESULTS

### Bacterial community composition based on sampling years, timing, and pasture management

There were differences in soil bacterial community structures at the phyla level between sampling years (2016 and 2017) and treatments (CG, H, RBR, and the control; $P < 0.05$). Community structure did not alter based on timing of sampling (pre or post poultry litter application; or April and July, respectively) ($P > 0.05$; Table 1 and Fig. 2). Although, there was a poultry litter application timing x year effect on bacterial community structure for 2016 and 2017. Based on PERMANOVA results, RBR and H communities differed at the phyla level between pre and post poultry litter applications, however there were no differences between pre and post applications for the CG and control treatments (Table 1; Fig. 2). OTU differences ($P < 0.05$) occurred at the genus level across systems, particularly following the long-term CG treatment and the control (Fig. S1).

Ten phyla dominated soil bacterial communities: Proteobacteria (mean relative abundance of all libraries was 30.8%), Actinobacteria (28.9%), Verrucomicrobia (8.8%),

**Table 1 PERMANOVA in bacterial community structure by years, timing (pre and post poultry litterapplication season), and pasture management.** PERMANOVA results illustrate differences in bacterial community structure by single factor of sample collection years (2016 and 2017), timing (pre and post poultry litter application season), and pasture management, as well as two factors (Year × Timing, Year × Pasture management, and timing × pasture management) and three factors (Year × Timing × Pasture management) in Booneville, AR from 2016-2017.

| Factor | Pseudo-F | P-value |
|---|---|---|
| Year | 7.70 | 0.001[*] |
| Timing (Pre-Post) | 1.61 | 0.161 |
| Pasture Management | 8.87 | 0.001[*] |
| Year × Timing | 2.88 | 0.028[*] |
| Year × Pasture Management | 2.51 | 0.008[*] |
| Timing × Pasture Management | 1.15 | 0.317 |
| Year × Timing × Pasture | 1.15 | 0.306 |
| | Year separation | |
| Factor | 2016 | 2017 |
| Timing (Pre-Post) | 3.31 ($P = 0.006$)[*] | 1.67 ($P = 0.15$) |
| Pasture Management | 4.38 ($P = 0.004$)[*] | 6.07 ($P = 0.001$)[*] |
| Timing × Pasture | 1.11 ($P = 0.307$) | 1.06 ($P = 0.389$) |

**Notes.**
[*]$P < 0.05$.

Firmicutes (7.1%), Bacteriodetes (6.4%), Planctomycetes (5.4%), Chloroflexi (4.3%), Gemmatimonadetes (2.9%), Saccharibacteria (1.5%), and Armatimonadetes (0.3%). Conservation pasture management influenced the four most abundant phyla; however, these differences were not consistent between years (Fig. 3). In both years, the relative abundance of Proteobacteria was higher in CG, H, and RBR treatments compared to the control in 2016 ($P = 0.01$) (Fig. 3A). In 2017, Proteobacteria had greater ($P = 0.026$) relative abundance in CG. The relative abundance of Actinobacteria did not differ between treatments in 2016 ($P = 0.33$) (Fig. 3B), but rather decreased in the CG treatment during 2017 (Fig. 3B). The relative abundance of Firmicutes did not differ ($P = 0.26$) between treatments in 2016, but was elevated ($P = 0.04$) in RBR in 2017 (Fig. 3C). The relative abundance of Verrucomicrobia (Fig. 3D) was greatest in the control for both years. The abundance of Proteobacteria and Verrucomicrobia was different among pasture management treatments in both sampling years ($P < 0.05$), while the relative abundance of Actinobacteria and Firmicutes was only different between pasture management treatments and the control in 2017.

## Bacterial community structure following 13-yrs of different pasture management treatments

PCoA of Bray–Curtis distance of the bacterial community structure visualized clustering differences between pasture management systems (Fig. 4). Due to the relative abundance of phylum differing based on pasture management, authors were then interested in identifying which taxa were most different between treatments. The LEfSe method was applied to identify which phylum were most discriminatory between the three pasture management systems and the control (Fig. S2) (*Segata et al., 2011*).

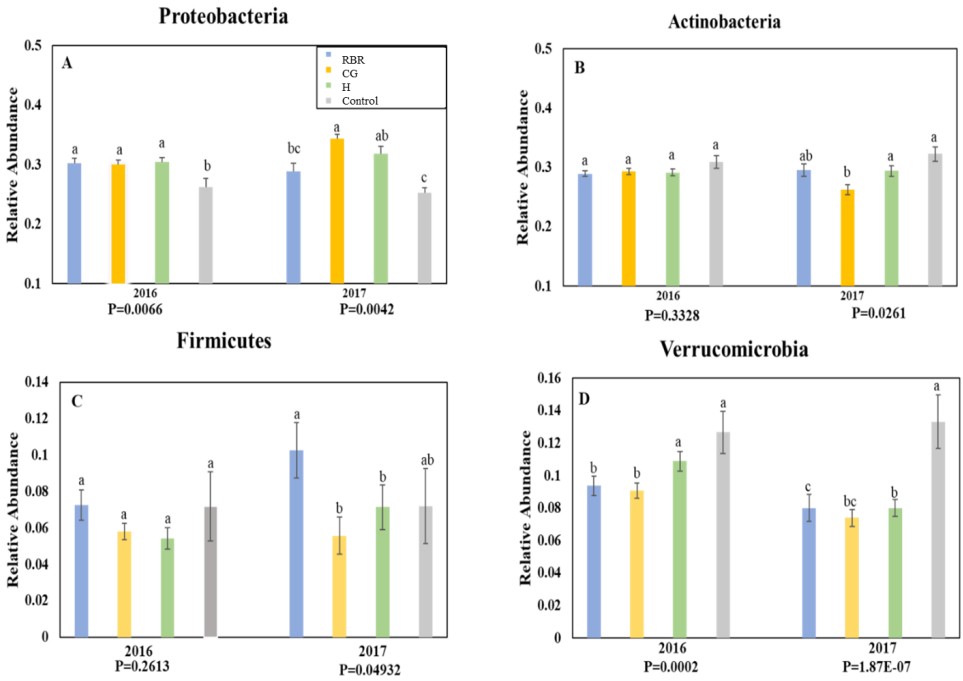

**Figure 3** **Relative abundance of bacteria phyla.** Proteobacteria (A), Actinobacteria (B), Firmicutes (C), and Verrucomicrobia (D) in years 2016 and 2017. Pasture management includes continuously grazed (CG), hayed (H), and rotational grazed with a fenced riparian buffer (RBR). The RBR treatment consists of an additional non-grazed zone without poultry litter and grazing (control). Blue = RBR, orange = O, green = H, and grey = control.

## Effect of sample location and poultry litter application timing effects on alpha diversity of bacterial communities

In both years, there were no effects of zone (or landscape position) on bacterial richness (Table 2). In addition, there were no effects ($P = 0.75$) of poultry litter application timing on bacterial richness in 2016 (pre: $\mu = 3,330.66$ and post: $\mu = 3,352.37$), while there was an increase ($P < 0.0001$) in richness during 2017 directly after litter applications (pre: $\mu = 3029.73$ and post: $\mu = 3,378.63$). Overall, pasture management effected bacterial richness in 2016, but not in 2017. When considering bacterial diversity, there was no effects of pasture management, sampling timing, or zone in either year. However, there were significant pasture management × timing interaction for diversity during both years (Table 2). Samples collected prior to annual poultry litter applications and under the H system had the lowest diversity ($\mu = 120.37$), whereas CG had greater bacterial diversity ($\mu = 153.17$; Fig. 5). Across all samples collected post poultry litter applications, CG again had higher diversity ($\mu = 155.45$), with RBR having the lowest ($P < 0.05$) microbiome diversity ($\mu = 130.28$). Therefore, following 13-years of poultry litter applications and management system implementation, diversity increased following poultry litter applications (CG: $\mu = 155.45$; H: $\mu = 144.98$; RBR: $\mu = 130.28$; control: $\mu = 131.61$) compared to pre poultry litter applications (CG: $\mu = 153.17$; H: $\mu = 120.37$; RBR: $\mu = 129.25$; control: $\mu = 120.64$).
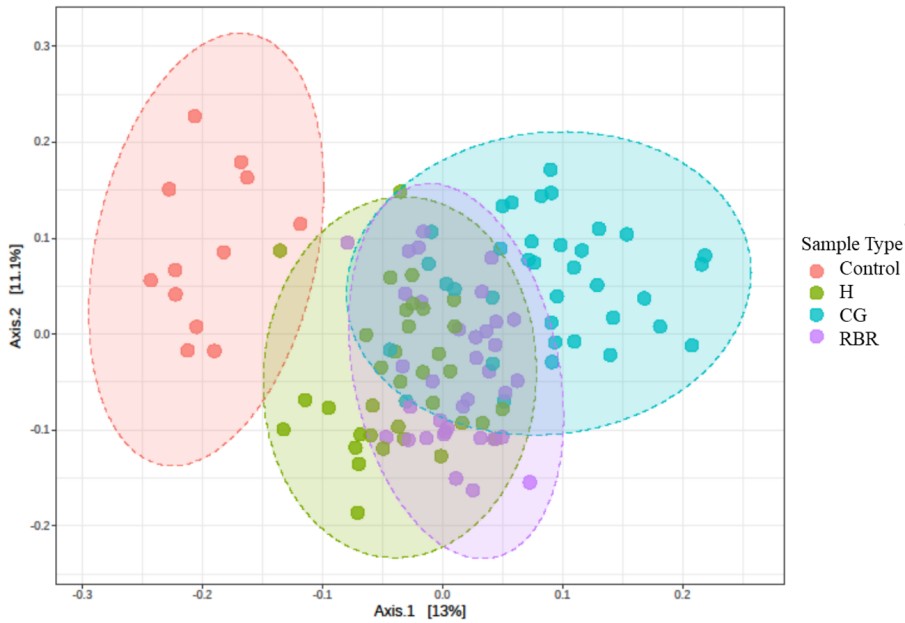

**Figure 4 PCA of bacterial community structure.** Principal Coordinated Analysis (PCoA) of Bray–Curtis distances of bacterial community structures in different pasture management. Pasture management includes continuously grazed (CG), hayed (H), and rotational grazed with a fenced riparian buffer (RBR). The RBR treatment consists of an additional non-grazed zone without poultry litter and grazing (control). Pink = control, green = H, blue = O, and purple = RBR.

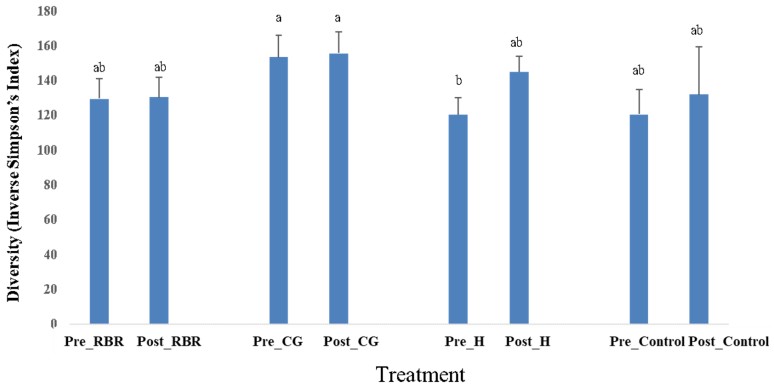

**Figure 5 Diversity in soil bacterial communities affected by pasture management × timing via inverse Simpson's index estimates cross years (2016 and 2017).** Pasture management treatments include continuously grazed (CG), hayed (H), and rotational grazed with a fenced riparian buffer (RBR). Pre and post indicate soil sampling before or after poultry litter was applied in these three treatments. The control is a non-grazed zone without poultry litter amendments at the downslope position of the RBR treatment.

## Bacterial community richness and diversity between pasture management treatments

Differences ($P = 0.05$; $F = 2.75$) occurred between treatments (Fig. 6), whereas no diversity differences occurred among pasture management treatments and the control (inverse

**Table 2  ANOVA of richness and diversity in bacterial community structure following 13-years ofpasture management, sampling date and zone.** ANOVA results illustrating richness and diversity in bacterial community structure by single factor of pasture management, sampling date and zone, as well as two factors (pasture management and timing) at Booneville, AR from 2016–2017.

| Parameter | Factor | df | F-value | P-value |
|---|---|---|---|---|
| Richness | 2016 | | | |
| | Pasture Management | 3 | 3.03 | 0.0367* |
| | Timing (Pre-Post) | 1 | 0.09 | 0.7583 |
| | Zone | 2 | 1.87 | 0.1642 |
| | Pasture × Timing | 3 | 0.45 | 0.7147 |
| | 2017 | | | |
| | Pasture Management | 3 | 1.30 | 0.2832 |
| | Timing (Pre-Post) | 1 | 17.55 | <0.0001* |
| | Zone | 2 | 0.21 | 0.8095 |
| | Pasture × Timing | 3 | 0.49 | 0.6876 |
| Diversity | 2016 | | | |
| | Pasture Management | 3 | 0.85 | 0.4679 |
| | Timing (Pre-Post) | 1 | 1.34 | 0.2501 |
| | Zone | 2 | 1.58 | 0.2146 |
| | Pasture × Timing | 3 | 4.16 | 0.0093* |
| | 2017 | | | |
| | Pasture Management | 3 | 0.79 | 0.5026 |
| | Timing (Pre-Post) | 1 | 1.04 | 0.3108 |
| | Zone | 2 | 1.80 | 0.1740 |
| | Pasture × Timing | 3 | 3.12 | 0.0318* |

Simpson's index) ($P = 0.09$; $F = 2.15$). Specifically, CG had greater soil community richness ($\mu = 3,405.6$) among all treatments followed by the control. In the present experiment, richness and diversity of soil bacterial assemblages in the control was lower ($\mu = 3,196.13$ and $\mu = 14.19$, respectively) than other treatments (H: $\mu = 15.33$; CG: $\mu = 14.32$; RBR: $\mu = 14.53$).

## Functional prediction and accuracy

We were interested in assessing potential shifts in nitrogen metabolism, as authors hypothesized that pasture systems receiving poultry litter and application timing would have different functional capacities with respect to N metabolism compared to the control. PICRUSt analysis identified 10 KEGG hits related to N metabolism. However, there were no distinct changes in potential N metabolism based on poultry litter applications, as we were unable to detect an influence of pasture management on N metabolism using this method. Considering the limitation of PICRUSt, biased primers and database limitation may result in inaccurate predictions (*Ashworth et al., 2017*). Additional functional diversity profiling analysis indicated no significant differences among treatments based on these KEGG metabolic functions.

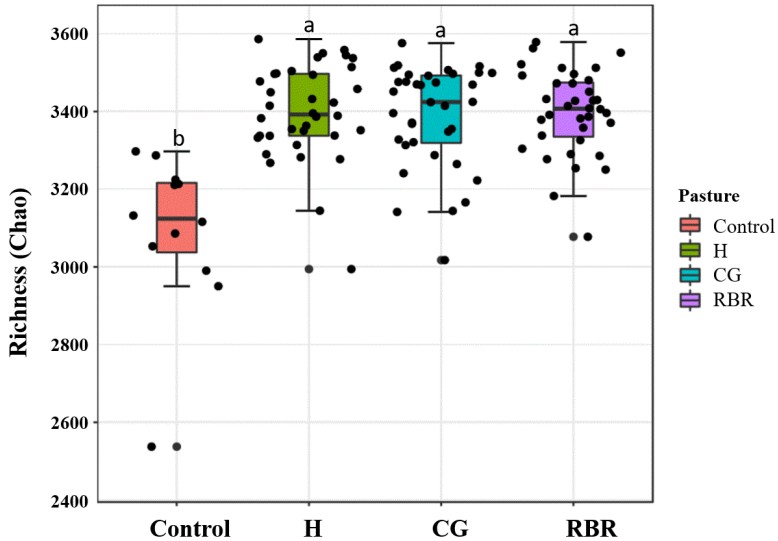

**Figure 6  Mean richness (Chao Estimate) in soil bacterial communities.** Pasture management include continuously grazed (CG), hayed (H), and rotational grazed with a fenced riparian buffer (RBR). The RBR treatment consists of an additional non-grazed zone without poultry litter and grazing (control).

## DISCUSSION

### Bacterial community composition based on sampling years, timing, and pasture management

Results illustrate the importance for evaluating soil ecology across time and space. Among the identified dominant phyla, Actinobacteria are affected by soil moisture; considering, Acidobacteria may decrease under greater precipitation and the presence of Proteobacteria increases (*Castro et al., 2010*). Consequently, greater (70%) humidity and soil moisture in 2017 corresponded with a lower relative abundance of Actinobacteria and a greater abundance of Proteobacteria. Verrucomicrobia was the third most abundant phyla detected in this survey; however, the ecology and soil functions of this phylum are not well understood. Although, Verrucomicrobia is reportedly the most dominant bacterial phylum in grasslands (*Bergmann et al., 2011*).

### Bacterial community structure following 13-yrs of different pasture management treatments

PCoA results revealed that soils receiving neither poultry litter nor cattle manure harbored distinct microbial communities compared to other treatments (Fig. 4). Overall, communities of the three pasture management systems (H, RBR, and CG) were most similar, likely owing to 13 continuous years of poultry litter applications compared to the control (no poultry litter). Previous research has also reported that long-term applications of poultry litter fundamentally drives soil bacteria community structure, due to a combination of altered soil physicochemical properties and poultry litter supplying it its own suite of microbes (*Ashworth et al., 2017*).

LEfSe results indicated that Verrucomicrobia and Abditibacterium (former phylum FBP) (*Tahon et al., 2018*) were the two most different taxa between treatments. Verrucomicrobia was very abundant in the control, while FBP was common in the CG treatment. Verrucomicrobia is ubiquitous in soil across a range of biomes in Antarctica, Europe, and the Americas and Verrucomicrobia was the dominant bacterial phylum in grasslands and in subsurface soil horizons (*Bergmann et al., 2011*). *Navarrete et al. (2015)* indicated that the relative proportion of Verrucomicrobia decreased after soil fertility increased following deforestation. However, the functions of both phyla in soil ecosystems are not well understood.

## Effect of sample location and poultry litter application timing effects on alpha diversity of bacterial communities

There were no effects of zone (or landscape position) on bacterial richness (Table 2). This suggests microbiome communities did not vary greatly across landscape positions, but rather long-term management overrides any terrain attribute impacts. Results indicates that poultry litter application timing influenced bacterial diversity in H systems, with no affect occurring on CG. One possible explanation for this is that cattle continuously graze in the CG system, thus continuously depositing cattle manure inputs (both nutrients and microbes), making this treatment less affected by poultry litter application timing. In the pre-poultry litter sampling date, H had the lowest bacterial diversity; however, the bacterial diversity in H increased following poultry litter applications. This indicates that poultry litter applications, particularly in H systems, results in more phylogenetically diverse communities owing to more favorable food sources in the rhizosphere of these forage crops and greater substrate for microbial metabolism.

One possible explanation of CG having greater diversity is that that stressed plants excrete compounds in the rhizosphere that provides substrates and encourages bacterial growth. In addition, as noted above, this result may be a direct effect of poultry litter, or it may also be due to greater soil temperatures at time of sampling for the post poultry litter application treatment ($\mu$ temperature $= 26.4\,°C$ in 2016 and $25.6\,°C$ in 2017) compared to the pre poultry litter application sampling period ($16.7\,°C$ in 2016 and $14.4\,°C$ in 2017). This finding also corresponds with results from *Ashworth et al. (2017)*, which observed lower richness and greater diversity when sampling under higher soil temperatures. Future research is needed to identify functional groups of soils amended with poultry litter in order to assess the potential for opportunistic pathogens (e.g., bacteroidetes) and identify potential functional shifts owing to manure deposition. Little research has been done on microbiome sequence differences between poultry litter and cattle manure following soil deposition, although sampling area, strategy, and location consistencies are needed for generating representative microbiomes for future microbiological-based manure studies. Considering, *Locatelli et al. (2017)* found that when poultry litter was sampled in-house (prior to land application), diversity (beta) and OUT abundance differed spatially. In a cattle feces community compositional study, *Wong et al. (2016)* observed temporal shifts in composition owing to moisture differences. Consequently, animal manure microbial

**Table 3 Soil chemical and physical results based on pasture management.** Soil sample analysis (0–15 cm) based on pasture management at Booneville, AR in 2016 and 2017. Pasture management includes continuously grazed (CG), hayed (H) and rotationally grazed treatment with a fenced riparian buffer (RBR). Poultry litter was applied in these three treatments. The RBR treatment consisted of an additional non-grazed zone without poultry litter amendments, which served as the control.

|  | Treatment | pH | P | K | Ca | Mg |
|---|---|---|---|---|---|---|
|  |  |  | Mg kg$^{-1}$ |  |  |  |
| 2016 | CG | 5.90 | 39.33 | 104.53 | 41.82 | 13.99 |
|  | H | 5.65 | 30.44 | 49.02 | 39.09 | 11.73 |
|  | RBR | 5.67 | 38.26 | 77.00 | 37.34 | 10.42 |
|  | Control | 5.71 | 5.01 | 39.13 | 25.90 | 5.93 |
| 2017 | CG | 5.48 | 54.72 | 212.74 | 55.28 | 21.77 |
|  | H | 4.95 | 39.56 | 52.13 | 45.52 | 13.74 |
|  | RBR | 4.99 | 44.04 | 109.88 | 45.16 | 12.91 |
|  | Control | 5.20 | 5.31 | 47.20 | 31.07 | 6.93 |

ecology varies widely based on environment, animal gut microbiome, sample type (fecal, litter), timing, and production practices.

## Bacterial community richness and diversity between pasture management treatments

Increases in grazing pressure increased soil communities, which was likely due to continuous manure inputs in the CG treatment. This result is similar to that of *Qu et al. (2016)*, which found that increased grazing increases soil bacterial community diversity. Cattle manure additions increase bacterial diversity, nutrient availability, aboveground plant biomass, and soil enzyme activity (*Das et al., 2017*). Increased microbial richness may be one reason that animal manure improves soil fertility and productivity in organic systems (*Kravchenko, Snapp & Robertson, 2017*). In the present experiment, richness and diversity of soil bacterial assemblages in the control was lower than other treatments, which suggests that cattle manure and poultry litter may be responsible for increasing soil diversity. *Ashworth et al. (2017)* also demonstrated that poultry litter applications increase diversity of soil bacterial communities. Conversely, some studies suggest grazing intensity decreases the soil microbial diversity (*Olivera et al., 2016*). Nonetheless, soil bacterial communities in agricultural soils are more temporally variable because of management and inputs when compared with other unmanaged ecosystems, such as grasslands and forest systems, which exhibit more seasonal stability (*Ashworth et al., 2017*; *DeBruyn et al., 2011*; *Lauber et al., 2013*). Nonetheless, study results highlight the importance of increased temporal and spatial sampling when evaluating soil microbes.

Many studies have established linkages between soil properties and soil microbiome (*Lauber et al., 2009*; *Rousk et al., 2010*; *Zhalnina et al., 2015*; *Fierer & Jackson, 2006*). The CG treatment had greater soil pH, P, K, Ca, and Mg compared to the control and other two pasture management systems (Table 3). In addition, nutrient concentrations in the control were lowest compared to CG, H and RBR, which is in accordance with lower species richness in control (Table 3). This suggests that soil systems management (such as

grazing management and animal manure inputs) may alter soil habitat by influencing the nutrient status and either stimulating or hindering microbial activity (*Balota et al., 2004*). In addition, soil health practices (e.g., conservation tillage, cover crops, crop rotation, and nutrient management; (*USDA-NRCS, 2019*) may increase soil microbial biomass, resulting in a larger pool of soil microbial biomass P (*Hallama et al., 2018*). As such, greater microbial diversity may increase the microbial biomass P pool and exacerbate P losses in runoff (*Turner & Haygarth, 2001*; *Blackwell et al., 2010*). Therefore, animal grazing density and nutrient management may drive soil biotic community structure and soil health across agricultural landscapes and these results can be used to identify best management practices for soil ecosystems.

## CONCLUSION

Poultry litter and cattle manure inputs increased soil bacterial diversity and richness, as well as altered the bacterial community composition in grasslands. In addition, these results suggest that microbiome communities do not vary greatly across landscape positions; rather management overrides impacts from terrain attributes. Richness differences found between long-term pasture management systems indicates that the number of species in the CG system was greater than H and RBR, all of which were greater than soils receiving no poultry litter or manure inputs (the control). Therefore, conservation agricultural practices (e.g., RBR) did not result in greater diversity, therefore, continuously grazed systems, albeit not recommended when water quality is a management consideration, did result in greater microbial diversity long-term. In addition, CG systems resulted in greater soil pH, P, K, Ca, and Mg, which corresponded with greater phylogenetic diversity. This outcome is reasonable given that poultry litter has a high abundance of bacteria, with $10^9$ CFU/g of aerobic bacteria (*Lu et al., 2003*). In addition, the high amount of nutrients in poultry litter are cofactors for bacterial growth and multiplication. Therefore, not surprisingly, the control (no poultry litter and cattle manure) had the lowest microbial diversity. Overall, these results highlight that animal inputs (both poultry litter and cattle manure) influence the soil pH and soil N, which can modify the makeup of soil microbial community and diversity by altering the nutrient status. Future research will focus on the presence of antimicrobial resistance genes in soils based on manure inputs.

## ACKNOWLEDGEMENTS

We appreciate Dr. Mary Savin (The Department of Crop, Soil and Environmental Sciences, University of Arkansas) for use of lab instruments. We also appreciate Taylor Cass Adams (Poultry Science Department, University of Arkansas) for transporting samples and preparing the experiments. Thanks for Jamie Hess (Poultry Science Department, University of Arkansas) for helping with soil sample preparation. Mention of tradenames or commercial products in this publication is solely for the purpose of providing specific information and does not imply recommendation or endorsement by the U.S. Department of Agriculture.

### Funding

This work was supported by the USDA Research Project: Conservation Pasture Management on Antimicrobial Resistance Bacteria in Water and Soil Systems: Longitudinal Analysis Over 13 Years. Project Number: 6022-63000-005-08-S. The funders had no role in study design, data collection and analysis, decision to publish, or preparation of the manuscript.

### Grant Disclosures

The following grant information was disclosed by the authors:
USDA Research Project: Conservation Pasture Management on Antimicrobial Resistance Bacteria in Water and Soil Systems: Longitudinal Analysis Over 13 Years: 6022-63000-005-08-S.

### Competing Interests

The authors declare there are no competing interests.

### Author Contributions

- Yichao Yang performed the experiments, analyzed the data, prepared figures and/or tables, approved the final draft.
- Amanda J. Ashworth conceived and designed the experiments, performed the experiments, analyzed the data, authored or reviewed drafts of the paper, approved the final draft.
- Jennifer M. DeBruyn analyzed the data, contributed reagents/materials/analysis tools, prepared figures and/or tables, approved the final draft.
- Cammy Willett approved the final draft.
- Lisa M. Durso contributed reagents/materials/analysis tools, prepared figures and/or tables, approved the final draft.
- Kim Cook contributed reagents/materials/analysis tools, approved the final draft.
- Philip A. Moore, Jr. conceived and designed the experiments.
- Phillip R. Owens approved the final draft, oversaw long-term managment of watersheds at this site.

### Data Availability

Data are available at Zenodo and Genbank:

Yang, Yichao, Ashworth, Amanda, Willett Cammy, DeBruyn, Jennifer, Durso, Lisa, Cook, Kim, …Owens, Philip. (2019). Soil bacterial biodiversity is driven by long-term pasture management, poultry litter, and cattle manure inputs [Data set]. Zenodo. http://doi.org/10.5281/zenodo.3333871

Genbank BioProject—PRJNA563928.

## Supplemental Information

Supplemental information for this article can be found online at http://dx.doi.org/10.7717/peerj.7839#supplemental-information.

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
