# Peer review of "Soil bacterial biodiversity is driven by long-term pasture management, poultry litter, and cattle manure inputs"

_PeerJ, doi:10.7717/peerj.7839_

## Round 0.1 · original submission · Major Revisions

The two reviewers found the manuscript interesting and suggested minor to major revisions. I think their comments can be addressed and if the manuscript is revised for format and style, I would recommend it for publication. I present some specific examples below but this is not a comprehensive list to guide these revisions.

Specific comments

Line 34. Understanding is not a component of soil health. Delete “understanding”
Line 65. I suggest using abberviations HS and RG to be consistent with CG.
Line 72. Here and throughout, avoid the passive voice. Delete “are able to”
Line 73. Avoid phrase like “some studies have suggested” unless you doubt those suggestions and intend to present an alternative hypothesis.
Line 76. Use previously defined abbreviation for rotational grazed.
Line 77. Delete phrase “are proven to”
Line 78. Replace “have been…reducing” with “reduces”
Line 79. Delete “Although .. communities”
Line 102. Delete “the”
Line 105. Replace “are among…affecting” with “control”
Line 107. Cite Fierer (2007) for soil pH correlation to diversity
Line 119. Replace “Through these…pasture management” with a statement of the main result of the work.
Line 174. Provide the similarity (97 % ?) used to define ribotypes in Mothur.
Line 475. Please proofread references. Present volume:pages but not issue and (line 501) only cap first letter of proper nouns in paper titles.

Reviewer 1 ·

Basic reporting

Please see my general comments below

Experimental design

Please see my general comments below

Validity of the findings

Please see my general comments below

Additional comments

In this manuscript, Yang et al described their study on the impact of long-term conservation pasture management on soil bacterial diversity using 16S rRNA gene amplicon sequencing. This is a very interesting studies. The manuscript is well written.

My comments are given below

1) The authors started with soil health. Based on the text, I can see microbiome diversity is an important factors. However, more detailed information on key factors affecting soil health is necessary for both introduction and discussion.

2) The raw data processing is quite ambiguous to this reviewers. It seems all analysis are at the phylum level. There is no mentioning of the basic unit OTU or ASV (amplicon sequence variant). Additionally, 16S data is often compared at family or genus level where functions are more specific or meaningful. Please explain here

3) Are there similar studies on soil microbiome using similar technology? In any case, it is important to start with summary of the overall community profiles at the different taxonomy levels before performing any comparison. This can serve as reference for future studies, for instance, a table or taxonomy trees describing the abundance of key taxonomies at different levels

4) Figure 5 LEFSe results is also strange. It should all more taxa with some of them are significant. Showing only two taxa is really unusual

Minor comments:

1) One important findings are the impact of poultry litter and cattle manure on microbiome diversity. This is clearly of agricultural interest by themselves. Are there studies on the microbiome of poultry litter or cattle manure? It will be very interesting to compare or at least comment on this matter

2) Figure 2 and 3 are a bit repetitive, could be merged to be more informative.

3) Suggest to remove: L324-336 and Supplementary materials are not very interesting as the predicting of microbiome functional potential are mainly for human gut microbiome, PiCRUST performance is very poor for soil micrcrobiome

4) L232-233 Please explain "these results suggest the main bacterial functions remained structural stable" the reviewer don't feel this is logical conclusion from the previous text

Reviewer 2 ·

Basic reporting

This paper meets the requirements for language, background, structure, and hypothesis.

Experimental design

The experiment consists of an appropriate, long-term trial and its research questions and methods are appropriate.

Validity of the findings

The paper suffers from only a need for minor corrections to clarify some findings.

Additional comments

In this paper, Yang et al. present their findings on the effects of a thirteen-year pasture management experiment on the soil microbiome. The authors show that land management and manure amendments created a gradient of soil bacterial richness which was lowest in sites that received no manure inputs and highest in continually-grazed cattle pastures. The authors suggest that increased grazing pressure may promote belowground biodiversity. The manuscript is generally clearly written, though there are a few minor points that need clarification (see below). My only major concern is the paragraph discussing PCoA based on Bray-Curtis dissimilarities across treatments. I feel this figure is being under utilized. PERMANOVA could also be applied to this section to determine if the clustering of the pasture management is meaningful. Otherwise, there are a few changes that I think are important for clarity, especially for readers coming from outside of the livestock/agronomy sphere.

Specific comments:

L40-Throughout: There is inconsistency in the use of broiler litter and poultry litter throughout the manuscript. I think it would be better to stick to poultry litter throughout.

L46-53: This summary is awkward as it alludes to the negative effects of grazing on plants, though this is not explored. Also these sentences are somewhat difficult to follow. The use of albeit to link disjointed statements does not help

L65-Throughout: As previously, there is inconsistency across the manuscript using H for hayed e.e. L117.

L75: Is penetration resistance referring to water or physical intrusion?

L89-90: Citation needed.

L96-97: This statement could benefit from further citation.
L103-104: Which citation goes with which statement?

L129-130: A citation for these soil classifications (i.e. FAO classification? USA classification?) is needed.

L132-149: A more clear description of the control treatment should go here.

L162-166: Were samples homogenized?

L180-183: I can’t tell if this is just a fancy way of saying rarefied?

L186: R is software, not a package.

L250-252: This discussion on Verrucomicrobia should be expanded. There is growing interest of its role in soils Navarrete et al 2015 (doi: 10.1007/s10482-015-0530-3), Bergmann et al., 2011 (doi: 10.1016/j.soilbio.2011.03.012), Cabello-Yeves et al., 2017 (doi: 10.3389/fmicb.2017.02131)

L256-throughout: The authors consistently remind the reader that they are using the Chao index. Personally I do not think this is needed, but the authors may disagree.

L314: Is microflora an appropriate term? Would microbiome be better?

Fig 5: I think this figure should be moved to the Supplementary Material as it does not really add anything to the paper.

Fig 7: Purely an aesthetic suggestion, but I think that the dots should be coloured by treatment as well to make the distribution of data clearer to

---

## Round 0.2 · Minor Revisions

The manuscript is improved and I believe acceptable for publication. I have some minor comments on style and format that the authors should consider. These follow from my previous comments.

Specific comments
Line 45. Consider replacing “greatest” and “greater” with “highest” and “higher”
Line 50. Avoid the passive voice. Replace “were effective at reducing” with “reduced” and “be a mechanism for improved” with “improve (line 53).”
Line 67. Avoid phrases like “is thought to” and “some studies have indicated that (line 82)” unless you doubt those reports and intend to present alternative hypotheses. I suggest revising to “This practice improves soil health and water quality and conserves renewable resources.” Also, did you mean nonrenewable resources?
Line 71. As above, the statement “are designed to” suggests that you have doubt that riparian buffer strips work. Also, I don’t think they prevent loss from soil but rather intercept nutrients in runoff before it reaches receiving waters. I suggest revising to “..strips decrease nutrient loading..”
Line 74. The phrase “reportedly leads to” suggests someone else has reported something else. Replace with “Overgrazing by livestock erodes soil…”
Line 75. Delete “whereas”
Line 80-83. Replace “includes…repeated applications” with “provides macronutrients and trace elements but repeated..”
Line 85. Phrases like “such as..manure” that can be deleted without changing the meaning of the sentence should be bracketed with commas.
Line 87-88. I don’t think we need to tell readers why bacteria, or life in general, need nutrients. Such a discussion is interesting in the context of Redfield ratios and the stoichiometry of nutrient assimilation but not here. Delete “Phosphorous..in bacterial cells.”
Line 94. I suggest replacing “microbiome plays an important role in providing support for plant growth” with “microbiome supports plant growth”
Line 96. Again, avoid passive voice. Revise to “whether grazing changes microbial function and diversity”
Line 108. Interesting choice of the word “confirmed” here. Did Fierer and Jackson confirm someone’s earlier work or did they discover that pH predicts bacterial diversity?
Line 214. Again, passive voice. I suggest “Ten phyla dominated soil bacterial communities..”
Line 231-236. This section belongs in Methods.
Line 262-266. “MicrobiomeAnalyst..compared to the control.” also belongs in Methods.
Line 267. Replace with “PICUSt analysis identified ten KEGG..”
Line 276-277. This is a weak start to Discussion. I suggest starting with a topic sentence and deleting “Proteobacteria and ….the relative abundance of”
Line 281. Suggest replacing “experiment” with “survey”
Line 288-289. Delete “on all the three…long-term applications of”
Line 296. Delete “widely considered to be”
Line 298. Delete “Research from”
Line 337. Delete “As it has been demonstrated that” and revise to “Cattle manure addition increases bacterial diversity, nutrient availability…”
Line 340. Delete “is recognized as an effective practice for” and replace “improving” with “improves”
Line 565. Please proofread references. Only cap proper nouns and the first word in journal titles.
Line 623. Revise to “Genome Biology” (cap both)
Line 628. Italicize genus and species.

---

## Round 0.3 · accepted · Accept

I appreciate your consideration of my suggestions.

Regards, Michael